# Evaluation of high-resolution meteorological data products using flux tower observations across Brazil

Jamie R. C. Brown[1], Ross Woods[1], Humberto Ribeiro da Rocha[2], Debora Regina Roberti[3], Rafael Rosolem[1,4]

[1]School of Civil Aerospace and Design Engineering, University of Bristol, Bristol, BS8 1TR, UK
[2]Instituto de Astronomia, Geofísica, e Ciências Atmosféricas, Universidade de São Paulo, São Paulo, 13083-8782, Brazil
[3]Departamento de Física, Universidade Federal de Santa Maria, Santa Maria, 97105-900, Brazil
[4] Cabot Institute for the Environment, University of Bristol, Bristol, BS8 1UH, UK

Correspondence to: Jamie Brown (Jamie.Brown@Hydrology.uk.com), Rafael Rosolem (Rafael.Rosolem@bristol.ac.uk)

**Abstract.** In the past decade, the scientific community has seen an increase in the number of global hydrometeorological products. This has been possible with efforts to push continental and global land surface modelling to hyper-resolution applications. As the resolution of these datasets increase, so does the need to compare their estimates against local in-situ measurements. This is particularly important for Brazil, whose large continental scale domain results in a wide range of climates and biomes. In this study, high-resolution (0.1 to 0.25 degrees) global and regional meteorological datasets are compared against flux tower observations at 11 sites across Brazil (for periods between 1999-2010), covering Brazil's main land cover types (tropical rainforest, woodland savanna, various croplands, and tropical dry forests). The purpose of the study is to assess the quality of four global reanalysis products [ERA5-Land, GLDAS2.0, GLDAS2.1, and MSWEPv2.2] and one regional gridded dataset developed from local interpolation of meteorological variables across the country [Brazilian National Meteorological Database (referred here as BNMD)]. The surface meteorological variables considered were precipitation, air temperature, wind speed, atmospheric pressure, downward shortwave and longwave radiation, and specific humidity. Data products were evaluated for their ability to reproduce the daily and monthly meteorological observations at flux towers. A ranking system for data products was developed based on the mean squared error (MSE). To identify the possible causes for these errors, further analysis was undertaken to determine the contributions of correlation, bias, and variation to the MSE. Results show that, for precipitation, MSWEP outperforms the other datasets at daily scales but at a monthly scale BNMD performs best. For all other variables, ERA5-Land achieved the best ranking (smallest) errors at the daily scale and averaged the best rank for all variables at the monthly scale. GLDAS2.0 performed least well at both temporal scales, however the newer version (GLDAS2.1) was an improvement of its older version for almost every variable. BNMD wind speed and GLDAS2.0 shortwave radiation outperformed the other datasets at a monthly scale. The largest contribution to the MSE at the daily scale for all datasets and variables was the correlation contribution whilst at the monthly scale it was the bias contribution. ERA5-Land is recommended when using multiple hydrometeorological variables to force land-surface models within Brazil.

# 1 Introduction

In regions that lack high-density meteorological monitoring networks or have sporadic historical observations, gridded weather products provide valuable historical references to aid studies for many purposes, including water resources (Syed et al., 2008; Vissa et al., 2019), flood forecasting and heatwaves (Miralles et al., 2019), prediction of vegetation dynamics and agricultural yields (Tian et al., 2019), and climate change impacts (Wagner et al., 2007; Dullaart et al., 2019; Terzago et al., 2020; Xi et al., 2021). These products provide a method to integrate available weather station data both temporally and spatially consistently, whilst taking into consideration factors of influence such as topography, prevailing winds, and distance (Thornton et al., 2021). They are becoming more readily available worldwide and are helping with regional to global applications where ground-based observations are not available or more consistent temporally extensive datasets are needed (e.g., Soti et al., 2010; Hughes and Slaughter, 2015; Gampe 2017). However, limitations in the forcing data can result in disinformation in data which can lead to incorrect conclusions (Beven, 2011; Kauffeldt et al., 2013) and therefore the validation of such products is essential to ensure a fair and reliable assessment of model performance. Comparison studies between these products and ground-based observations over the study area is one way to validate and determine its reliability and suitability.

New efforts are being made to validate global data products for important hydrological applications. For example, Sikder et al. (2019) tested three GLDAS versions and ERA-Interim/Land products over South and Southeast Asia (the Ganges-Brahmaputra-Meghna and Mekong River basins) against discharge observations to determine which product better describes the system. Gebrechorkos et al. (2020) used rainfall observations to analyse the ability of two gridded high-resolution datasets to detect climate variability and droughts across East Africa, whilst Weber et al. (2021) compared multiple gridded products against an Alpine observation centre to determine their capability for snow hydrological modelling.

The selection of a gridded product is based on its suitability for long-term hydrological applications, which require consistent meteorological forcing data spanning over 20 years or more. However, it is also important to compare products in data-sparse areas whenever ground-based observations are available to provide insight on the local-to-regional uncertainties associated with the product. Higher uncertainties potentially affect the ability to prepare for climate events by local or regional institutions. Furthermore, increased data in areas with scarce availability strengthens model representation of Earth system processes (IPCC, 2012).

With centers such as the European Centre for Medium Range Forecasts (ECMWF), and the National Aeronautics and Space Administration (NASA) Goddard Space Flight Center (GSFC)'s Global Modelling and Assimilation Office (GMAO) producing openly available high-resolution global gridded products using different techniques it can be difficult to know which products may be better suited for each application. Different products may excel in some areas over others due to the nature of their interpolation/reanalysis method and the ground observations used. For example, products like the ECMWF's ERA5-Land (Muñoz-Sabater, 2019) are developed from blending observations with past short-range weather forecasts rerun with modern weather forecasting models to produce many land-surface flux variables. MSWEPV2.2 (Beck et al. 2019), however, focuses only on precipitation data and combines satellite remote sensing data with multiple sources of reanalysis products,

then bias corrects and weights between multiple nearby observation gauges. Multiple studies have been undertaken showing that different products provide contrasting results depending on the environment or climate in question (Decker et al., 2012; Wang and Zeng, 2012; Sikder et al., 2019; Beck et al., 2021).

The global distribution of weather stations tends to be biased toward populated areas leaving large areas underrepresented (Viana et al., 2021). For example, in 2018 study that utilised 7,759 rain gauges across Brazil revealed that the Amazon basin,

which contains 70% of the country's freshwater, has the lowest density of gauges, with only 724 (<10%) in the entire basin with Brazilian borders (combined states of Acre, Amapá, Amazonas, Pará, Rondônia, Roraima, and Mato Grosso) (Filho et al., 2018). Places of ecological importance, such as large forests and savannahs, like the Amazon and Cerrado, influence the hydrological cycle through a variety of factors including; biodiversity, vegetation dynamics, and root distribution (Oliveira et al., 2005; Diaz et al., 2007, Bonal et al., 2016, Coe et al., 2016) despite low meteorological station density. To address this,

meteorological observation data from measurement stations such as flux towers provide valuable insight into representative areas across the country.

FLUXNET is an international network of flux towers where eddy covariance techniques are used to measure energy, water and carbon fluxes between the biosphere and atmosphere (Baldocchi et al. 2001), however, their distribution is highly biased towards North America and Europe. This network of towers has provided opportunities to validate gridded products (e.g.

reanalysis products) over regions with dense observational coverage, such as North America (Decker et al., 2012) and China (Wang and Zeng, 2012). However, comparatively less work has been undertaken in regions with limited observational coverage, such as South America, although there have been some efforts to compare evapotranspiration products (derived from land surface models, reanalysis, and remote sensing) (Sörensson and Ruscica, 2018; Gomis-Cebolla, 2019; Andrade et al., 2024). The tower sites in this study are part of FLUXNET but no attempt thus far has been made to comprehensively evaluate

gridded products using these locations. In this study, the evaluation is centred on core meteorological variables (Table 2 and Table 3), as these represent the fundamental hydrometeorological drivers used in land surface modelling and hydrological applications. Flux-derived variables such as evapotranspiration or latent heat flux are not considered in this study, but remain an important avenue for future work.

Here the accuracy of one regional and four global high-resolution gridded meteorological products are compared over 11 ecologically diverse flux tower sites spanning multiple biomes across Brazil (Fig. 1, Table 1a and b). Specifically, the study aims to answer four questions. Firstly, which high-resolution gridded product is the most accurate overall when compared with local observation data? Secondly, which product demonstrates the highest accuracy for each variable when evaluated against observational data? Thirdly, what are the dominant types of error associated with each product when compared to observation

data? And finally, how do these errors vary spatially and seasonally?

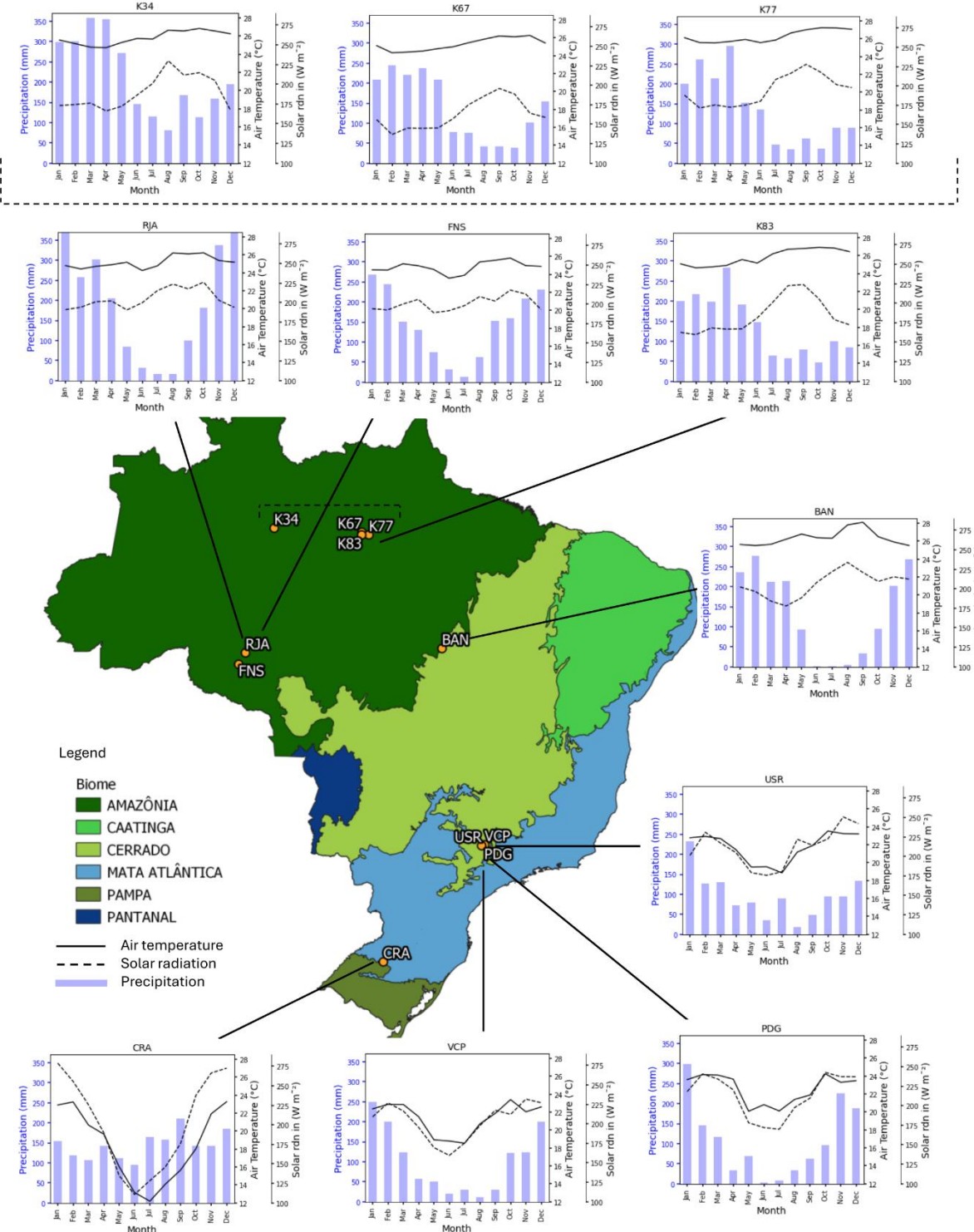

**Figure 1: Map of Brazil spilt into its six major biomes showing the location of the study sites analysed in this paper and their measured average monthly precipitation, air temperature, and incoming irradiance.**

**Table 1a: Summary of the abbreviated names, lat, lon, location, and period of observation, for the 11 flux tower sites analysed in this paper. All sites were provided directly by the principal investigator. ***

| Name | Lat (°N) | Lon (°E) | Location (State) | Start | End |
|------|----------|----------|------------------|-------|-----|
| K34 | -2.60908 | -60.2093 | Manaus (AM) | 1999 | 2006 |
| K67 | -2.85667 | -54.9589 | Tapajos (PA) | 2002 | 2006 |
| K77 | -3.0119 | -54.5365 | Tapajos (PA) | 2000 | 2005 |
| K83 | -3.017 | -54.9707 | Tapajos (PA) | 2000 | 2004 |
| CRA | -28.6034 | -53.6736 | Cruz Alta (RS) | 2009 | 2014 |
| FNS | -10.774 | -62.3374 | Ji-Parana (RO) | 1999 | 2002 |
| RJA | -10.0832 | -61.9309 | Jaru (RO) | 1999 | 2002 |
| PDG | -21.6206 | -47.63 | Luis Antonio (SP) | 2001 | 2003 |
| VCP | -21.5833 | -47.602 | Ribeirao Preto (SP) | 2005 | 2009 |
| BAN | -9.82442 | -50.1591 | Araguaia (TO) | 2003 | 2006 |
| USR | -21.6371 | -47.7903 | Luis Antonio (SP) | 2005 | 2008 |

**Table 1b: Summary of the abbreviated names, vegetation type, elevation (abs), average temperature (T), average monthly precipitation (precip) and references for the 11 flux tower sites analysed in this paper.**

| Name | Land Cover | Elevation (m) | Mean T (°C) | Mean Monthly Precip (mm) | Reference |
|------|-----------|---------------|-------------|--------------------------|-----------|
| K34 | Tropical Rainforest | 130 | 25.9 | 206 | Araujo et al. 2002 |
| K67 | Tropical Rainforest | 88 | 25.3 | 140 | Rice et al. 2004 |
| K77 | Cropland - Pasture | 130 | 26.3 | 137 | Sakai et al. 2004 |
| K83 | Tropical Rainforest | 153 | 25.9 | 125 | Goulden et al. 2004 |
| CRA | Cropland (soybean) | 432 | 18.3 | 144 | Webler et al. 2012 |
| FNS | Cropland - Pasture | 306 | 24.8 | 138 | Kirkman et al. 2002 |
| RJA | Tropical Dry Forest | 145 | 25.3 | 166 | von Randow et al. 2004 |
| PDG | Savanna | 690 | 22.6 | 107 | Rocha et al. 2002 |
| VCP | Cropland (Eucalyptus) | 761 | 21.3 | 102 | Cabral et al. 2010 |
| BAN | Woodland Savanna | 120 | 26.3 | 136 | Borma et al. 2009 |
| USR | Cropland - Sugarcane | 552 | 21.6 | 94 | Cabral et al. 2013 |

*The site names are abbreviations used throughout this paper. K stands for Kilometer followed by a number which is the name of the access road to the site within the Amazon. The other sites are abbreviations also regarding their location listed here: CRA – Cruz Alta, FNS – Fazenda Nossa Senhora, RJA – Rebio Jaru, PDG – Pé de Gigante, VCP – Votorantim, Fazenda Cara Preta, BAN – Bananal Island, USR – Usina Santa Rita

## 2 Datasets

### 2.1 In-situ observations

Meteorological data were obtained from the 11 sites across Brazil (Fig. 1, Table 1a and b). These sites represent those for which continuous, multi-year, quality-controlled meteorological forcing data were available, including all variables required for the calculation of reference evapotranspiration ($ET_0$) following FAO methodology (Allen et al., 1998). Although additional towers exist, such as CAX (Caxiuanã, a tropical rainforest riverine site in the state of Pará) and USE (Usina Santa Eliza, a sugarcane site in the state of São Paulo), they were excluded from this study due to insufficient coverage or data quality issues

that prevented them from meeting the thresholds required for analysis. The emphasis of this study is on the fundamental meteorological drivers themselves, rather than on derived fluxes such as $ET_0$, which are model-dependent quantities and addressed further in Section 5.5.

  The selected towers provide representation of Brazil's largest biomes, particularly the Amazon and Cerrado, as well as croplands and grasslands/pastures. However, some biomes, notably the Caatinga and Pantanal, are not represented in the

present dataset due to the absence of suitable flux tower data at the time of carrying out this study. This omission is acknowledged as a limitation.

  Conventional meteorological stations from the Instituto Nacional de Meteorologia (INMET) were not used in this study. Importantly, some of the reanalysis and blended products assessed (e.g., BNMD) incorporate INMET station data in their development. Using flux tower data, which are independent of INMET, allows for a more objective evaluation of these

products. This approach aligns with established practice; for example, FLUXNET data have been used in a similar manner for the evaluation of MSWEP precipitation datasets because they remain fully independent of the products under assessment (Beck et al., 2022).

  Data from the flux tower sites cover periods ranging from four to seven years, although not always overlapping. Variables include air temperature, precipitation, wind speed, air pressure, longwave and shortwave radiation, and specific humidity

(Table 2 and Table 3). These are the primary hydrometeorological drivers used in land surface models and they underpin the estimation of surface energy and water fluxes, allowing for the calculation of evapotranspiration, which in turn provides essential inputs for many hydrological models. Where necessary, variables were converted to ensure uniformity and comparability across sites, and additional variables (e.g., specific humidity) were derived from recorded parameters such as vapour pressure, dew point temperature, and relative humidity.



**Table 2** Availability of each variable for each flux tower site. The meteorological variables are; *ws,* wind speed, *ta,* air temperature, *press,* atmospheric pressure, *rgs,* short wave radiation, *par,* photosynthetically active radiation (used to calculate *rgs*), *rgl,* long wave radiation, *prec,* precipitation, *ee,* vapour pressure, *dpt,* dew point temperature, and *RH*, relative humidity. *ee, dpt* and *RH* variables were used to calculate specific humidity.

| Site | ws | ta | press | rgs | par | rgl | prec | ee | dpt | RH |
|------|----|----|-------|-----|-----|-----|------|----|-----|----|
| | | | | | Meteorological variables | | | | | |
| K34 | X | X | X | X | | X | X | X | | |
| K67 | X | X | X | | X | | X | X | | |
| K77 | X | X | X | X | | X | X | | | X |
| K83 | X | X | X | X | | X | X | X | | |
| CRA | X | X | X | X | | | X | | X | |
| FNS | X | X | X | X | | X | X | X | | |
| RJA | X | X | X | X | | X | X | X | | |
| PDG | X | X | X | X | | | X | X | | |
| VCP | X | X | X | X | | | X | | | X |
| BAN | X | X | X | X | | | X | X | | |
| USR | X | X | X | X | | | X | | | X |


## 2.2 Regional Products

Efforts have been made to produce high-resolution datasets through interpolation of weather stations (Xavier et al., 2016). The meteorological station network across Brazil varies spatially and temporally with few data available before 1980. There has
been a steady increase in weather stations and rain gauges over the last 40 years but with heavy bias towards stations closer to densely populated areas such as São Paulo and Rio de Janeiro (Alvares et al., 2013; Filho et al., 2018). These biases bring the quality of meteorological datasets under scrutiny and a strong need for validation especially over the more data poor areas. Gridded data products were selected based on their open access availability, a spatial resolution of 0.25 x 0.25 degrees or finer, and a daily or sub-daily temporal resolution (Table 3) referred to here on as high-resolution.

**2.2.1 Brazilian National Meteorological Gridded Database (BNMD)**

A high-resolution gridded dataset developed from local interpolation of meteorological variables across Brazil was released in 2016 spanning 1980-2013 (0.25 by 0.25 degrees, daily) (the Brazilian National Meteorological Database, referred here as BNMD) (Xavier et al. 2016). The data were collected from 3625 rain gauges and 735 weather stations over this period and quality control procedures were performed to identify outliers based on Liebmann and Allured (2005). A lack of previous

reviews due to the novelty of this dataset combined with the rapid increase in stations/gauges over the 30-year period of data that has been made available, raises questions about its reliability, particularly over less data rich areas such as the Amazon.

### 2.2.2 Global Land Data Assimilation System (GLDAS) 2.0 and 2.1

In 2004, NASA-GSFC and NCEP released a reanalysis data product called Global Land Data Assimilation System (GLDAS) (Rodell et al. 2004). Since then, GLDAS has been reprocessed leading to the updated release of GLDAS2.0 in November 2019
and GLDAS2.1 in January 2020. GLDAS2.0 data are products of the new NOAH-3.6 LSM forced using the Princeton meteorological forcing dataset (Sheffield et al. 2006) producing a dataset from 1948 – 2014. GLDAS2.1 is a direct update from GLDAS-1 where NOAH-3.6 LSM is forced with combined forcing data including Global Precipitation Climatology Project (GPCP) version 1.3 produced by NOAA with available data from 2000-present (both datasets 0.25 by 0.25 degrees, 3-hourly).

### 2.2.3 ECMWF Reanalysis 5-Land (ERA5-Land)

In 2019, the European Centre for Medium Weather Forecasting (ECMWF) released ERA5-Land (an upgraded form of ERA-Interim) providing a higher resolution global land-based dataset from 1981-present (2025) (0.1 x 0.1 degrees, hourly) (Muñoz-Sabater, 2019) generated using Copernicus Climate Change Service Information. The production of ERA5-Land is the result of the tiled ECMWF Scheme for Surface Exchanges over Land incorporating land surface hydrology (H-TESSEL). The recent
release sees it benefit from over a decade of developments in 4D-VAR data assimilation, core dynamics, and model physics relative to GLDAS and ERA-Interim. As it integrates a wide array of global observation data sources, employs advanced data assimilation techniques, and benefits from continuous improvements the quality would be expected to be higher than that of new regional datasets such as ones produced by Xavier et al. (2016).

### 2.2.4 Multi-Source Weighted-Ensemble Precipitation v2.2 (MSWEPv2.2)

Another recent dataset that garnered significant attention is the Multi-Source Weighted-Ensemble Precipitation, version 2.2 (MSWEP V2.2) (Beck, et al., 2019). Although only precipitation data, it provides high temporal (3-hourly) and spatial (0.1 degrees) resolution based on gauges, satellites, and reanalysis with distributional bias corrections. The dataset merges multiple observation, satellite and reanalysis data across the globe and its predecessors have proven to provide reliable estimates for precipitation patterns globally dating from 1979-2017 (Baez-Villanueva et al. 2018; Moreira et al. 2018; Alijanian et al. 2019;
Xu et al. 2019).

Table 3 provides a summary of the gridded products used in this study with information about time periods covered, temporal and spatial resolution, the meteorological variables accessed and their references.


**Table 3 Summary of gridded data products used in this study**[*]

* Refer to Table 2 for abbreviations. q refers to specific humidity. Note: temporal resolution was converted to the most coarse of the datasets – Daily. Monthly values were also generated.

| Data Descriptor | Data Source | Variables accessed | Periods | Temp. res. | Spatial res. | Reference |
|---|---|---|---|---|---|---|
| BNMD | BNMD | Prec, maxTa, minTa, rg, press, ws, RH | 1980-2017 | Daily | 0.25x0.25 | Xavier et al. 2016 |
| ERA5-Land | ERA5-Land | Prec, Ta, rg, rgl, press, ws, dpt | 1981-2019 | Hourly | 0.1x0.1 | Muñoz Sabater, 2019 |
| GLDAS2.0 | GLDAS_NOAH25_3H 2.0 | Prec, Ta, rg, rgl, press, ws, q | 1948-2014 | 3-hourly | 0.25x0.25 | Rodell et al. 2004 |
| GLDAS2.1 | GLDAS_NOAH25_3H 2.1 | Prec, Ta, rg, rgl, press, ws, q | 2000-2019 | 3-hourly | 0.25x0.25 | Rodell et al. 2004 |
| MSWEPv2.2 | MSWEP_v2.2_sh | Prec | 1979-2017 | 3-hourly | 0.1x0.1 | Beck et al. 2019 |

## 3. Methodology

This section describes the data manipulation necessary that enabled us quantification of the differences between the gridded products and observations.

### 3.1 Quality control

Flux tower data frequently contain gaps or periods with suboptimal data quality due to a variety of reasons (e.g. sensor malfunction, drifting, calibration errors, power supply issues). To address this, the selection of variables for each site was
guided by their data availability, consistency, and a requirement for completeness, with more than 80% data coverage achieved after infilling (Jung et al., 2024). Furthermore they were analysed for trends to identify potential measurement drifts caused by instrumentation. Initial quality control and gap-filling procedures had already been conducted by the principle site investigator. Despite this, remaining errors related to faulty instrumentation were removed from the dataset. Gaps in the seven meteorological variables were subsequently filled using linear regression, prioritising variables from the same site that
exhibited strong correlations ($R^2 > 0.8$), with the most robust correlations being utilised first. Instruments logging similar measurements were primarily used for gap filling. For example, most stations measured wind speed using both eddy covariance techniques and an anemometer, logging almost identical measurements, yet overlapped different periods of time. Similarly, there were multiple instruments measuring temperature, shortwave and longwave radiation, and humidity.

### 3.2 Temporal averaging

Flux tower sites have different recording methods and temporal resolutions. All observation and gridded datasets were converted to the coarsest common temporal resolution, the daily scale, for analysis (BNMD, Table 3). As the gridded datasets have no gaps this was a straightforward forward or backward averaging depending on the variable and averaging method. Two-sample Kolmogorov-Smirnov (K-S) tests were carried out on the observation data for each variable, where full days (24-hours) were used to create daily data and set as the reference distribution. Samples were then tested against this distribution

using one less hour each iteration to determine whether samples significantly deviated from the reference sample. A minimum of 12 hours of data (50% available) was set as the threshold for daily conversion. This choice was based on a sensitivity analysis in which stricter thresholds (100, 90, 80, 80, 70, 60%) were tested. While higher thresholds led to a reduction in the number of valid days across sites, gains in accuracy were marginal when evaluated against the reference mean and standard deviation. The 50% threshold therefore represented a pragmatic balance between representativeness and data availability. Although systematic biases cannot be fully excluded (e.g., from gaps clustering during specific conditions), the analysis showed that daily and monthly estimates were not significantly affected.

A similar averaging method was adopted to convert daily data to monthly. However, due to a shortage of data availability, instead of using 100% of days available in a month as the reference sample, 80% or above was used to acquire a more representative sample to test against. Evaluation of a lower inclusion threshold (50% of days in a month) demonstrated that monthly means and standard deviations remained consistent, supporting its use as a minimum conversion criterion.

Precipitation was summed when converting to daily and monthly. Rainfall does not follow a regular pattern or known distribution, meaning taking anything less than all 24 hours of available data would result in an under-prediction. Therefore, only days with all hourly data available were converted to daily. The same approach was taken converting daily to monthly but, in some cases, resulted in a high loss of data. To conserve data, each site was assessed uniquely looking at the two-sample K-S test results and changes in the mean after using fewer days in the month to convert (i.e., rejected if changes in the means and standard deviations were >2%). A scaling factor was then applied to the monthly total depending on how the percentage of days missing to bring the total up to 100%.

### 3.3 Wind Speed vertical interpolation

The height at which the measurement instruments are located differ at each site. To compare data products to the observation data they are vertically interpolated to the height of the instrument at each site. The BNMD wind speed variable was calculated by interpolating laterally from the nearest Brazilian weather station which records wind speed over grass. The ERA5-Land 10 m wind speed product is produced for comparison against surface synoptic observation (SYNOP) stations, also above grass. GLDAS 10 m wind speed is adjusted down from the model's lowest level to 10m but it is unclear whether this is over grass or different vegetation types. For consistency, the same vertical interpolation method was used for all data products. The wind speeds were interpolated up the log-wind profile using grass as the vegetation type at the height of the WMO weather station standard (30 cm) from either 2 m or 10 m depending on the data product.

### 3.4 Atmospheric Pressure

Atmospheric pressure is not provided as a variable in the BNMD dataset; thus, it was estimated as a single continuous value following the method outlined in the FAO Irrigation and Drainage Paper No. 56 (based on data homogeneity and availability). Incorporating this estimated value allowed for a critical comparison of its performance relative to other measured variables,

providing insight into whether the observed variables performed better or worse than a single mean estimate. The atmospheric pressure variable was available for comparison in the ERA5-Land and both GLDAS products.

### 3.5 Specific humidity

Specific humidity was available for both GLDAS datasets but needed to be calculated for ERA5-Land and BNMD. ERA5-Land water vapour pressure was calculated from the dew point temperature variable and then converted to specific humidity using pressure (Shuttleworth, 2012). For BNMD, the vapour pressure at maximum and minimum temperatures was calculated using the FAO method (Eq. 11, Allen et al. 1998). These were then used with the relative humidity, and estimated pressure to calculate specific humidity using ideal gas laws (Bolton, 1980).

### 3.6 Decomposition of the mean square error

To quantify differences across variables, datasets, and temporal scales, the Mean Square Error (MSE) was calculated for each variable - data product combination relative to the observational data at each site. The MSE is a single metric with limited explanation about the source of the error, but it can be decomposed into parts to acquire a better understanding of contributions to the error (Gupta et al. 2009) (Eq. 1)).

$$MSE = 2 \cdot \sigma_s \cdot \sigma_o \cdot (1 - r) + (\sigma_s - \sigma_o)^2 + (\mu_s - \mu_o)^2 \tag{1}$$

In Eq. 1, $\sigma_s$ and $\sigma_o$ are the standard deviations of the sample (gridded product) and observations, $r$ is the linear correlation between the sample and the observations, and $\mu_s$ and $\mu_o$ are the means of the sample and observations respectively. Written like this the equation is seen to have three parts. The first term is the correlation contribution to the MSE, the second, the variation contribution or differences in standard deviation, and the third term represents the bias contribution or differences in means. For a clearer visualisation of the results, the individual error source to MSE was scaled to the RMSE magnitude to conserve the units for each variable (Iwema et al., 2017). Whilst the MSE is beneficial when comparing products, quantifying the relative contributions to the MSE provides valuable insights into the reasons behind discrepancies between the observations and the gridded products.

### 3.7 Performance ranking

The large volume of data, spanning five gridded products, decades of time series, and seven variables across multiple observation sites, presents significant challenges in summarising the results into a coherent structure. For this reason, a ranking system was used to ascertain which data product for each variable performed best due to its simplicity (Brunke et al. 2003). The MSE for each variable was given a rank dependant on how many data products have that variable recorded (for incoming longwave radiation this was 1-3; precipitation, 1-5; and all other variables 1-4, 1 being the best performing/lowest MSE). This was carried out for each site and then the ranks were averaged across all 11 sites to provide a single rank for each variable and

each data product. A product with the lowest MSE for a variable over all 11 sites would score a rank of 1. An overall average rank was then also given to each data product which only included ranks of variables that were present for all products. This method of ranking was performed for both the daily and monthly data.

**3.8 Sensitivity to observational record length**

To assess the influence of dataset length on performance metrics, a sensitivity analysis was carried out using subsets of the longest flux tower records (K34, K77, CRA, and VCP). For each site, shorter records of 2 and 4 years were extracted and compared against the full observational record. Errors were evaluated as the percentage difference relative to the observational mean (RMSE / $\mu_o$), and the maximum and minimum values across all gridded products were calculated. The analysis showed

that all subset errors deviated by less than 10% from those obtained using the full observational record. This indicates that performance metrics are robust to record length within the range tested. Full results are presented in the supplementary material (Fig. S3-S6).

**4 Results**

In this section, precipitation and air temperature are analysed separately due to their fundamental importance in hydrological

and climatological studies. These two variables exert significant influence on ecosystem dynamics and are the more widely used in model validation and environmental monitoring. A more detailed examination is thus warranted. The remaining variables, while important, are discussed collectively for brevity, as they primarily serve to complement the analysis of temperature and precipitation.

The methods outlined in Section 3 were applied to five products at 11 observation sites for seven meteorological variables, at

295 both daily and monthly timescales. An example of the results is shown in Fig. 2, which illustrates monthly air temperature at the BAN site. The visual representation of the MSE components allows for a clearer interpretation of the performance of each dataset. In Fig. 2a, the partial contributions to the MSE are colour-coded and plotted as stacked columns, where the total column height reflects the total MSE, scaled to the RMSE. The column height differences across products facilitate direct comparison of their performance.

To further understand these results, Fig. 2b offers additional context, illustrating the key sources of error presented in Fig. 2a. For example, the large bias contributions for the BNMD and GLDAS2.0 datasets are attributed to consistent overpredictions of observed data with higher mean temperatures. Despite this, the seasonality of both datasets aligns closely with observations, resulting in a lower contribution from variability errors. In contrast, the GLDAS2.1 dataset displays higher variability error, likely due to overprediction during the hot months and underprediction during the cooler periods, though it has a smaller bias

contribution due to a similar overall mean with the observation data. The ranking scores in Table 4 complement these

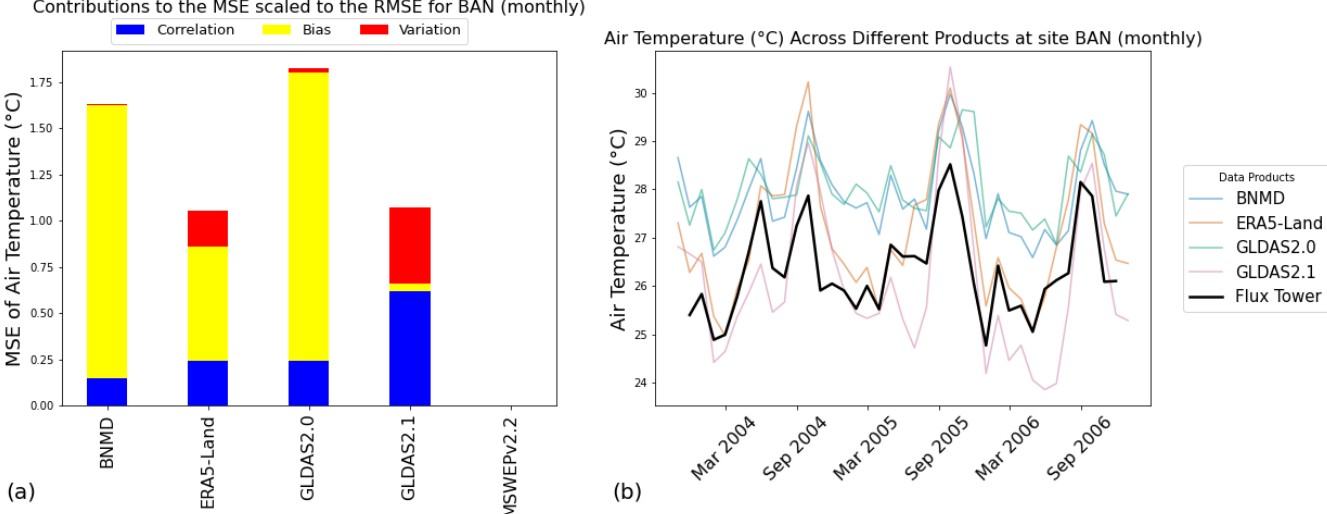

**Figure 2 Air temperature at site BAN, (a) partial contributions to the MSE for each gridded dataset, and (b) the monthly time series of all gridded datasets for air temperature over the operation period of the flux tower (observation data in bold).**

visualisations (Fig. 3 and Fig. 4) by providing a comprehensive comparison of the MSE for each dataset and variable, averaged over the 11 flux tower sites and presented alongside an overall ranking across all variables.

## 4.1 Precipitation

Precipitation data from all five gridded datasets were statistically analysed at both daily and monthly temporal resolutions, allowing for a comprehensive comparison across scales. MSWEPv2.2 consistently performed the best at the daily scale, while GLDAS2.0 exhibited the poorest performance at both daily and monthly resolutions. However, when considering monthly precipitation, BNMD demonstrated the best overall performance among the compared products (see Section 5.2).

Figure 3 and 4 illustrate the partial contributions to the MSE for each dataset at daily (Fig. 3) and monthly (Fig. 4) scales, providing insight into the sources of error across sites. At the daily scale (Fig. 3), the primary contributor to the MSE is the correlation error, particularly at sites such as RJA and K34, where variability errors are also prominent. Interestingly, bias contributes very little to the total MSE on a daily basis. However, this pattern shifts at the monthly scale (Fig. 4), where bias plays a much larger role. For instance, in the northern Amazonian sites (K67, K77, and K83), the ERA5-Land dataset consistently overpredicts monthly rainfall, resulting in large spikes in the bias component of the MSE (see Section 5.3).

Another key observation is the variability contribution, which is notable high at sites RJA and USR. While most datasets capture the total rainfall during dry months, they tend to overpredict rainfall during the wet season, increasing the variability error. Although MSWEPv2.2 generally outperforms the other datasets, it did not perform consistently across all sites. For example, at the daily scale, MSWEPv2.2 had the poorest performance at K83, where timing discrepancies during the wet season led to high correlation errors. Similarly, at the monthly scale, MSWEPv2.2 performed worst at BAN, where overpredictions during peak wet months resulted in higher bias and variability contributions to the MSE.

## 4.2 Air Temperature

Air temperature was analysed across four of the gridded datasets. ERA5-Land performed best whilst GLDAS2.0 performed least well at both daily and monthly scales (Table 4). The ranking of 1 and 1.09 indicates that ERA5-Land had the lowest MSE when compared with every other dataset across all sites at the monthly scale and all sites except one (CRA) at the daily scale respectively. The monthly BNMD dataset performed equally as poorly as GLDAS2.0 meaning the ranking system is unable to identify which dataset reflects the in-situ observations least well. Correlation error has the largest contribution to the MSE

for the daily datasets with the bias contribution also having some influence (Fig. 3). However, as found with precipitation, at the monthly scale, the greatest source of error shifts across all datasets and sites from the correlation contribution to the bias contribution (Fig. 4).

**Table 4 Overall ranks for MSE. MSE is taken per variable per site and ranked (Table S1-S12). Ranks are then averaged for all sites to produce an overall rank for daily and monthly data. Both the lowest (bold) value (i.e. best performance) and highest (italics) value**
**(i.e., worst performance) in each row are identified. Dashed cells (-) indicate no data available. Ranking for precipitation incorporates the fifth dataset MSWEPv2.2 in its calculation.**

| | Daily | | | | |
|---|---|---|---|---|---|
| | BNMD | ERA5-Land | GLDAS2.0 | GLDAS2.1 | MSWEPv2.2 |
| Wind Speed (m s$^{-1}$) | 2.55 | **1.64** | *3.27* | 2.55 | - |
| Air Temperature (°C) | 3.36 | **1.09** | *3.64* | 1.91 | - |
| Pressure (hPa) | 2.18 | **1.82** | 2.82 | *3.18* | - |
| Shortwave rdn in (W m$^{-2}$) | *3.55* | **1.36** | 2.36 | 2.73 | - |
| Longwave rdn in (W m$^{-2}$) | - | **1** | 2.2 | *2.8* | - |
| Precipitation (mm) | 2.04 | 2.62 | *3.42* | 2.33 | **1.6** |
| Specific Humidity (kg kg$^{-1}$) | *4* | **1.18** | 2.45 | 2.36 | - |
| Average Rank (exc. Longwave) | 2.95 | **1.62** | *2.99* | 2.51 | - |

| | Monthly | | | | |
|---|---|---|---|---|---|
| Wind Speed (m s$^{-1}$) | **1.91** | 2 | *3.36* | 2.73 | - |
| Air Temperature (°C) | *3.27* | **1** | *3.27* | 2.45 | - |
| Pressure (hPa) | 2.18 | **1.82** | 2.82 | *3.18* | - |
| Shortwave rdn in (W m$^{-2}$) | 2.73 | *3* | **1.91** | 2.36 | - |
| Longwave rdn in (W m$^{-2}$) | - | **1** | *2.6* | 2.4 | - |
| Precipitation (mm) | **1.6** | 3.13 | *3.2* | 2.26 | 1.82 |
| Specific Humidity (kg kg$^{-1}$) | *4* | **1.27** | 2.36 | 2.36 | - |
| Average Rank (exc. Longwave) | 2.62 | **2.04** | *2.82* | 2.56 | - |

Both the BNMD and GLDAS2.0 datasets consistently overpredict air temperature explaining the bias contributions for both monthly and daily datasets. Although performing well overall, the monthly GLDAS2.1 dataset had the largest variability contributions which are explained by overpredicting temperatures in the hotter months and underpredicting them in the cooler

months (K34, K67, K77 & BAN). The ERA5-Land dataset followed the mean of the observation data most closely but varied in either overpredicting or underpredicting temperature at different sites.

## 4.3 Other Meteorological Variables

Besides precipitation and temperature, five other meteorological variables were analysed: wind speed, pressure, downward shortwave and longwave radiation fluxes, and specific humidity. Among these, wind speed exhibited the poorest performance

in GLDAS2.0 across both temporal resolutions, whereas ERA5-Land demonstrated the highest accuracy at the daily scale and BNMD at the monthly scale. Substantial bias errors are evident at site FNS across all datasets, with gridded datasets underpredicting observation data by means ranging 45-75%. At site CRA, a high degree of variability and bias contributes to the MSE, with datasets underestimating observed values from 2009 to 2013. However, from 2013 to 2014, the observed wind speed declines uncharacteristically, leading to an overestimation by the gridded products.

ERA5-Land proved to have the lowest MSE on average whilst GLDAS2.1 performed least well at both temporal resolutions when analysing pressure. It is worth noting that the ranking did not change between daily and monthly scales for pressure as performance consistency was unaffected between daily and monthly datasets. The errors associated with pressure are heavily dominated by the bias contribution (Fig. 3 and Fig. 4). Contributions to the variability error are visible for BNMD as pressure was estimated using the elevation of the site using the standard FAO method (Allen et al. 1998) and kept as a constant figure.

BNMD's relatively low MSE when compared to other datasets tells us that estimating a single value for pressure can sometimes more accurately reflect the observation data. The large consistent biases at sites K67 and RJA are due to an overprediction and underprediction of 23 hPa and 19 hPa on average, respectively, for all datasets.

ERA5-Land performed best again at the daily scale for the variable, downward shortwave radiation whilst BNMD performed least well. Surprisingly, ERA5-Land performed least well at the monthly scale while GLDAS2.0 performed best. The

correlation contribution to the MSE dominated across all sites at both daily and monthly scales, but as temporal resolution decreases, so does the correlation contribution resulting in lower overall MSEs. Large bias contributions are evident at site K67 over both temporal resolutions as the gridded datasets consistently overpredict the observation data by around 40 W m$^{-2}$. The observation data tends to have a downwards trend over the entire recording period resulting in an increased bias towards the end of the time series when comparing to the gridded products.

Only three of the gridded data products and five sites recorded measurements of downward longwave radiation leading to its exclusion in the overall ranking across all variables in Table 4. ERA5-Land performed best at all sites across both time scales whilst GLDAS2.1 and 2.0 performed least well at the daily and monthly scales, respectively. All datasets tend to underpredict at every site, with contributions to all three components of the MSE visible at both time scales. However, the scale of the errors is not large, ranging between 1-7% error across the spread of the data.

With regards to specific humidity, ERA5-Land outperformed the other gridded datasets again whilst BNMD had the weakest performance at both time scales. BNMD's large biases are due to the estimation of vapour pressure from the minimum and maximum temperatures and a constant estimate for pressure (see Section 3.4). Biases associated with air temperature for

BNMD can therefore be expected to be seen in specific humidity. Similarly, the variability contributions to BNMD MSEs found at sites PDG, CRA, VCP and USR are associated with the variability errors in pressure as this was also utilised in the calculation.

The ranks were averaged for all shared variables for BNMD, ERA5-Land, GLDAS2.0 and GLDAS2.1 across both time scales. ERA5-Land performed best on average whilst GLDAS2.0 performed least well at both monthly and daily scales.

## 4.4 Seasonality in Errors

Errors throughout the year can change if the datasets fail to capture the correct range of seasonality. For example, dry seasons may have low errors in precipitation because the mean rainfall will be closer to 0. Figure 5 shows this behaviour across almost all sites for the best performing precipitation dataset, MSWEPv2.2. Similarly, biases may occur if datasets overpredict temperatures in the warmer seasons as seen in Fig. 6 at sites, BAN and FNS. It is clear from Fig. 1 that seasonality changes with latitude and that sites located further south have a higher range of temperature between seasons. This increased seasonality helps explains the relatively large errors seen at sites CRA and USR in Fig. 6. Comparing the errors spread over of the year between datasets helps us determine which ones best predict the seasonality. For example, take BNMD air temperature, the correlation component's contribution to the MSE increases in the summer months the more southerly the site, suggesting there is a weakness in the datasets ability to predict seasons. Further graphical representations of this can be found in the supplementary material.

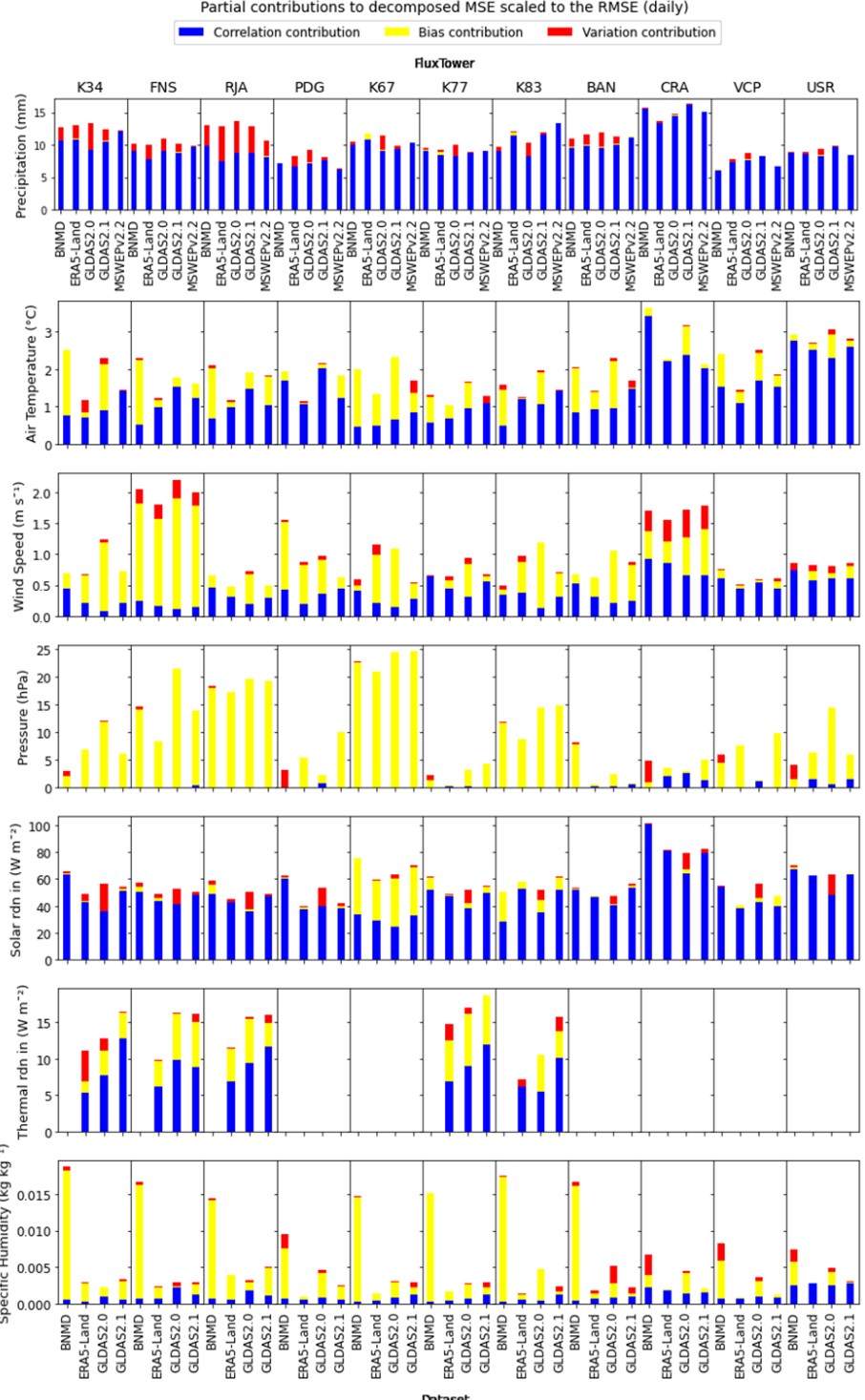

**Figure 3 Partial contributions to the MSE between each observation site and gridded dataset (x-axis) for each variable (y-axis) at the daily scale. Precipitation includes MSWEPv2.2 errors.**

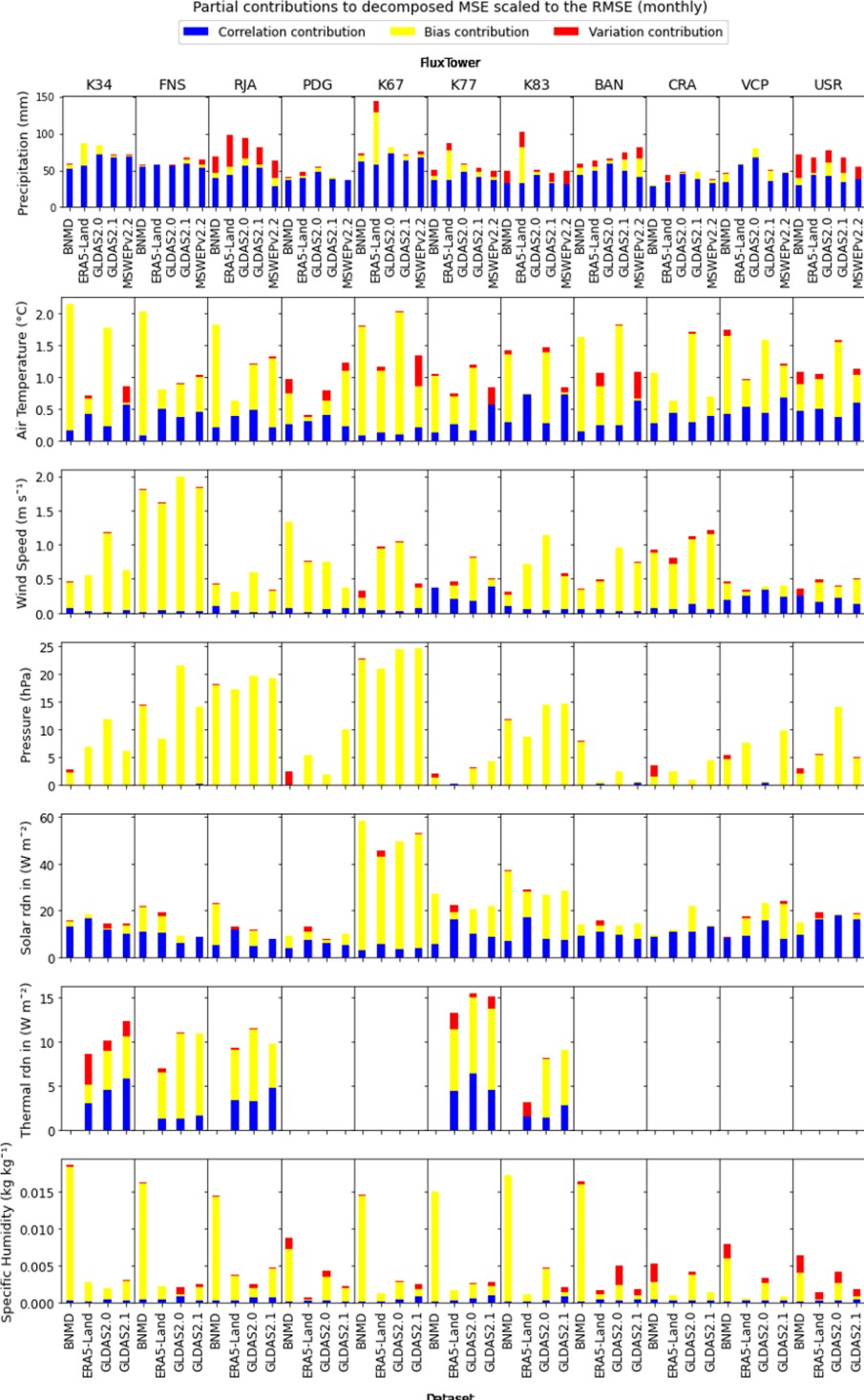

**Figure 4 Partial contributions to the MSE between each observation site and gridded dataset (x-axis) for each variable (y-axis) at the monthly scale. Precipitation includes MSWEPv2.2 errors.**

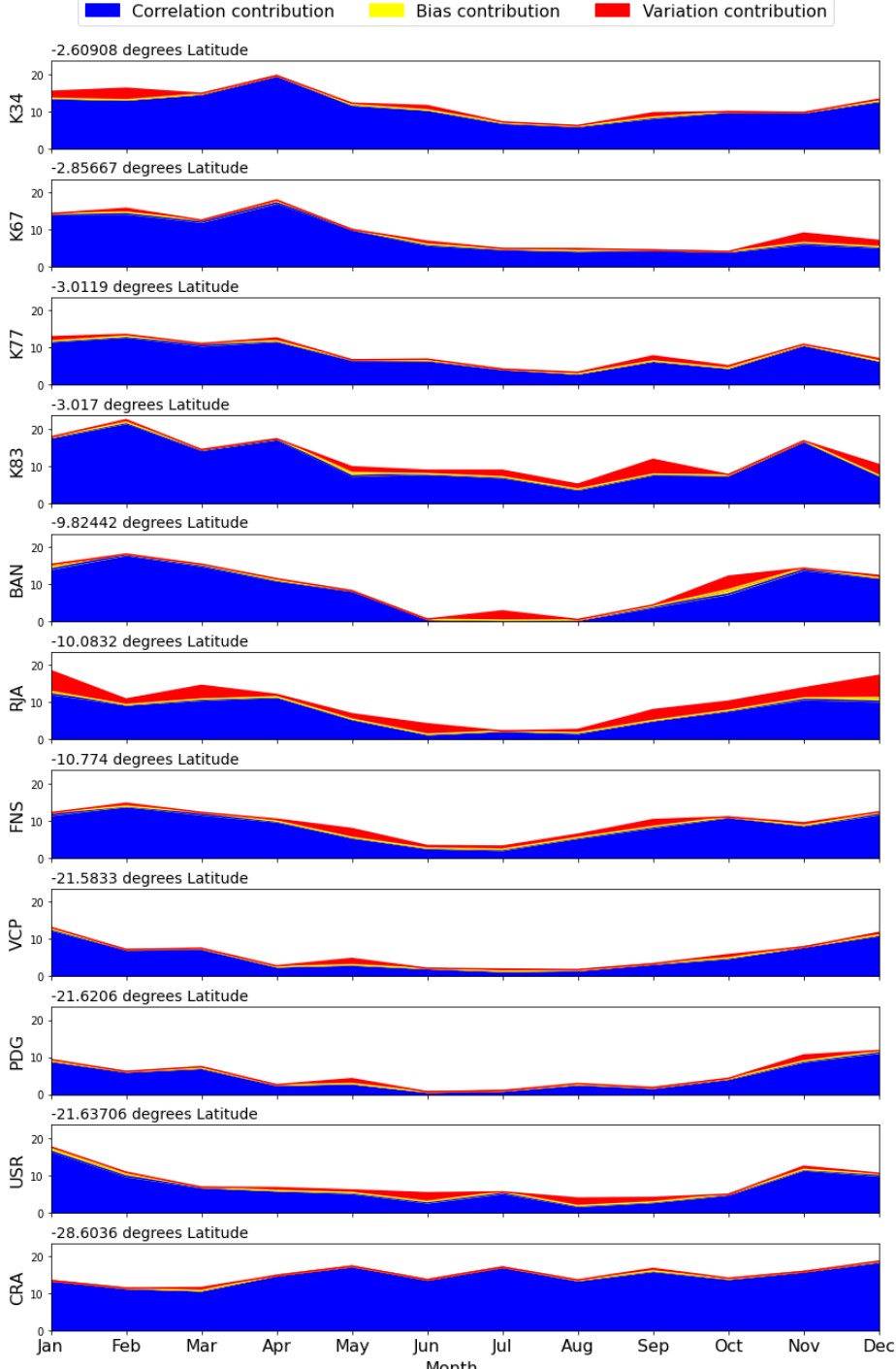

**Figure 5 Partial contributions to the MSE averaged by month over all operational observation years for MSWEPv2.2 precipitation across all sites. Sites are in descending order from distance from equator.**

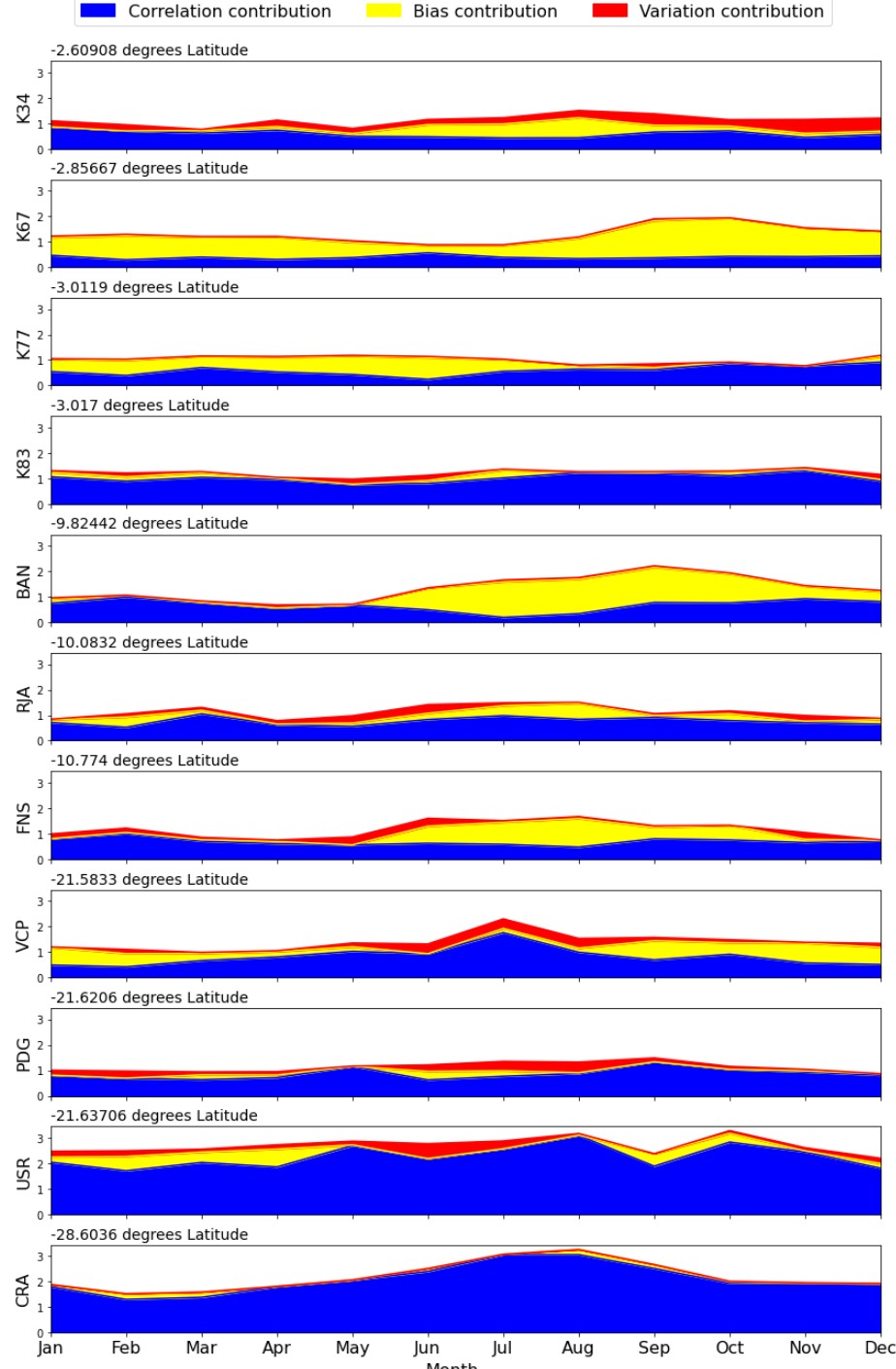

**Figure 6 Partial contributions to the MSE averaged by month over all operational observation years for ERA5-Land air temperature across all sites. Sites are in descending order from distance from equator.**

## 5. Discussion

Five gridded data products (BNMD, GLDAS2.0, GLDAS2.1, ERA5-Land, and MSWEPv2.2) were evaluated with in situ measured meteorological variables across multiple biomes in Brazil. The products were evaluated against 11 flux tower stations for seven meteorological variables (air temperature, wind speed, pressure, downward shortwave and longwave radiation, and specific humidity). Stations are spread over a variety of different Brazilian climates, and daily and monthly observational averages (or totals) were compared against the gridded products. The MSE scaled to the RMSE was calculated and intercompared among different products using a ranking system. Three additional statistical metrics (the correlation contribution, the variance contribution, and the bias contribution) were also computed to provide further insight into the cause of error.

### 5.1 Recommendations for overall product

It was found that ERA5-Land performs best overall for representing multiple meteorological variables at both daily and monthly scales. This finding is consistent with studies such as Decker et al. (2012), and Wang and Zeng (2012) which indicated that ERA-Interim (ERA5-Land's predecessor) generally outperformed other datasets when validated against 33 North American flux towers and 63 China Meteorological Administration (CMA) weather observation stations over the Tibetan Plateau, respectively. Similar results were reported by Jiang et al. (2020), Pelosi et al. (2020) and Zandler et al. (2020), who noted that ERA5's advanced spatial and temporal resolution contributed to superior representation of meteorological conditions. However, this study confirms that no single dataset consistently outperforms others across all variables or time scales, aligning with the conclusions of Decker et al. (2012) and Wang and Zeng (2012). Therefore, the importance of regional validation on global products is underscored.

### 5.2 Recommendations for each variable

At the daily scale, ERA5-Land was found to be the most accurate for all variables except precipitation, where MSWEPv2.2 aligned more closely with observations. This finding aligns with multiple other studies which demonstrated that MSWEP exhibited strong precipitation representation, particularly in data-scarce regions (Alijanian et al. 2019; Xu et al. 2019), including South America (Moreira et al. 2019). At the monthly scale, analysis shows that ERA5-Land best represents pressure, air temperature, longwave radiation, and specific humidity, while BNMD performs best for wind speed and precipitation. Comparisons can be drawn between this study and Decker et al. (2012), who despite comparing previous versions of some of the products analysed here (ERA-Interim, ERA-40 and GLDAS1.0), they concluded ERA-Interim (the predecessor of ERA5/ERA5-Land) outperformed the other products across most variables. As mentioned above, ERA-Interim also performed well when compared to CMA measurements across the Tibetan Plateau (Wang and Zeng, 2012). The dataset achieved the best performance at both daily and monthly air temperatures whilst also demonstrating low biases and high correlations in other

variables, such as precipitation. Given that ERA-Interim is the predecessor of ERA5-Land, with many parallel techniques used in dataset production, similar performance can be expected in this study.

To provide a clearer understanding of the absolute deviations between observations and best-performing product, a time series comparison for air temperature and windspeed at site PDG are presented in Fig. 7. This example offers a visual interpretation of a small section of Fig. 3 while addressing the need for absolute metric comparisons. The time series reveals that ERA5-Land consistently underestimates wind speed consistently by approximately 0.5 m/s, while showing only minor deviations in air temperature. Such performance is reasonable for a gridded product, highlighting its strength in capturing seasonal trends.

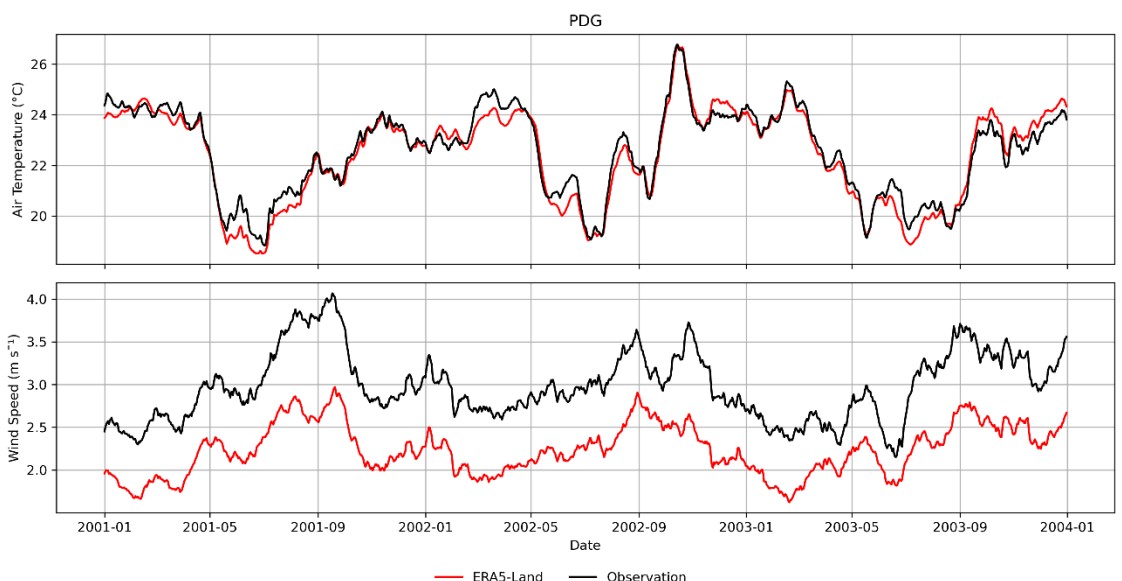

**Figure 7 The 30-day rolling average for air temperature and wind speed at site PDG for observation measurements (black) and ERA5-Land (red)**

Surprisingly, however, GLDAS (version 1) performed best in both daily and monthly precipitation in Wang and Zeng's 2012 study, while its successor, GLDAS2.0, ranked worst at both time scales in this analysis. This discrepancy implies that different datasets may not perform consistently on a global scale and may exhibit superiority over others depending on regional and climatic contexts.

The outperformance of BNMD over MSWEPv2.2 when the temporal resolution becomes coarser, suggests that despite

capturing daily patterns less well, BNMD captures the overall seasonality better than MSWEPv2.2. One explanation could be that BNMD has a greater correlation error but lower errors in variation and bias at the daily scale. When both datasets are "smoothed out" with the decrease in temporal resolution, BNMD's correlation error drops the overall MSE more than MSWEP. Both products use an extensive network of rain gauges to create the gridded product, but use different methods of interpolation (as well as inevitably a few different sources). It is with interest that they outperform each other at different scales as this

proves that different approaches to the creation of data products could prove suitable depending on the concerning temporal scale.

Meanwhile, GLDAS2.0 outperformed other datasets only for downward shortwave radiation at the monthly scale, consistent with results from Decker et al. (2012) and Wang and Zeng (2012), where GLDAS (version 1) excelled in solar radiation on other continents. The GLDAS2.0 product is forced using the Global Meteorological Forcing Dataset from Princeton University, which combines incoming shortwave radiation from NCEP reanalysis (Kalnay et al., 1996) and NASA Langley surface radiation budget data (Cox et al., 2017), using monthly data. This could explain GLDAS2.0's superiority at the monthly scale, as the monthly signal from the forcing datasets is effectively conserved.

## 5.3 Dominant types of error

Analysis reveals that the dominant sources of error vary significantly depending on the variable and time scale, reflecting the complexity of accurately capturing different meteorological factors (Fig. 3 and Fig. 4). At finer time scales (e.g., daily), correlation error emerges as the largest contribution for most variables, including precipitation, air temperature, solar radiation, and thermal radiation. This is likely because these variables fluctuate quickly over time, making it challenging for models to maintain alignment with observed temporal patterns. Conversely, errors associated with bias contribute the most to pressure and specific humidity, likely due to the stable nature of these variables and the potential accumulation of systematic offsets. Wind speed displays a more balanced distribution of error contributions from both correlation and bias, possibly reflecting the variable's high sensitivity to local topographic and atmospheric conditions, which can vary across sites. This finding mirrored observations by Decker et al. (2012) who acknowledged that the correlation contribution was more prominent at the daily and sub-daily scale.

When shifting to a coarser time scale (e.g., monthly), the dominant error contributions shift as well. Specifically, the relative contribution of bias increases across all variables, while correlation contributions decrease aligning with Decker et al. (2012) who also noticed this shift. This is anticipated, as temporal averaging at a monthly scale reduces the impact of time lags inherent in correlation errors, effectively smoothing out short-term discrepancies and highlighting systematic biases. Such changes underscore the importance of temporal resolution in model evaluations, as daily errors may understate or overstate the importance of correlation and bias depending on the analysis period.

For instance, the bias associated with wind speed might stem from assumptions made during vertical interpolation across datasets. In particular, assumptions about atmospheric stability or wind profile shape could introduce systematic errors that manifest as bias, especially at coarser time scales. Pressure, being relatively stable, may not capture short-term fluctuations as errors but instead reveal a tendency for overprediction or underprediction that surfaces as a steady bias. Specific humidity, which depends on pressure, is similarly prone to bias errors due to its sensitivity to any pressure-related inaccuracies.

These findings highlight the need to consider both temporal scale and variable characteristics in future model development and error correction approaches. Such detailed breakdowns offer a clearer understanding of the nuances in model performance, which could guide targeted improvements for specific variables and time scales.

## 5.4 Variation in error by location and seasonality

Results demonstrate a clear seasonal component in the errors for precipitation (e.g. Fig. 5, see supplementary Fig. S1) and shortwave radiation (see supplementary Fig. S2), with lower errors during dry seasons and heightened errors during wetter season. Precipitation error is expected as there is higher chance for error with more rain. Solar radiation error could be associated with increased cloud cover as it follows a similar pattern and are more difficult to replicate in modelled systems such as LSMs.

Speculating on the latitudinal impact on error proves challenging, as no clear patterns emerge. This does not imply that latitude lacks influence; rather, other factors, such as the dominant vegetation type, may obscure potential trends. Notably, the correlation contribution in air temperature does appear to be affected, with errors generally increasing with distance from the equator. It is acknowledged that larger sample sizes typically yield more robust correlations; however, in this study the number and distribution of flux tower sites are constrained by data availability. As such, while sample size may play a role, the focus is on evaluating the relative performance of gridded products against independent observations, rather than quantifying the effect of sample size on correlation strength but is an intriguing avenue for further investigation.

## 5.5 Methodological and instrument limitations

While analysis incorporated quality-controlled observational data, inherent limitations in flux tower measurements, such as instrument errors and episodic operation, remain a concern. As highlighted by Hollinger and Richardson (2005), flux tower data can deteriorate over time, which may introduce discrepancies when comparing these observations to gridded products, and in this study all data that was inside the scope of the variability, and therefore could not be rejected, was kept.

In addition to these instrument-related issues, the spatial coverage of flux tower data across Brazil is itself a constraint. The 11 sites included in this study represent those for which complete, high-quality data were available, covering important biomes (namely the Amazon and Cerrado) both natural and agricultural. However, some biomes, notably the Caatinga and Pantanal, are not represented in the present dataset due to the absence of suitable flux tower data. This omission is acknowledged as a limitation of the study, while highlighting an important direction for future work.

Conventional meteorological stations from INMET were not incorporated for similar reasons. At least one of the gridded products evaluated (BNMD) already assimilates INMET station data. Using these stations would have reduced the independence of our evaluation. By relying on flux tower data, which remain independent of the gridded products, the study provides a more objective benchmark. Nevertheless, this decision reduced the number of available sites and may have limited representativeness.

Additionally, reanalysis and interpolation methods differ among datasets, introducing unique biases. For instance, MSWEP and BNMD utilise distinct approaches to rain gauge data interpolation, resulting in varied precipitation accuracy depending on the region and scale. BNMD dataset is based on the spatial interpolation of a network of meteorological stations, generally installed at a height of 1.5 m in a standard WMO grass-covered area (Xavier et al., 2016). Vertical interpolation of air

temperature was not undertaken due to the complexity of modelling sub-canopy temperature gradients and the absence of sufficient high-resolution vertical profile data. As a result, temperature discrepancies may have arisen, particularly with BNMD, where air temperatures at ground level were likely higher than those at forest canopy height, contributing to the observed overestimation.

Another limitation arises from the mismatch between point-based flux tower observations and grid-cell values from coarse-resolution products. While such mismatches can influence point-to-pixel comparisons, particularly at daily time scales, spatial harmonisation through interpolation or downscaling was not applied. Previous studies have shown that interpolation accuracy is highly variable and often lacks consistent geographic patterns. For example, Hofstra et al. (2008) demonstrated this for European climate datasets, and Xavier et al. (2016) reported similar challenges when evaluating interpolation methods across Brazil. These findings indicate that harmonisation may not systematically improve agreement with observations and could introduce additional biases. For this reason, the focus of this study was placed on evaluating the temporal performance of gridded against independent, high-quality point observations, acknowledging the spatial representativeness remains a constraint.

Moreover, the spatial resolution of datasets appears to limit accuracy at daily time steps, where smaller-scale variability is more critical. However, this limitation becomes less prominent at monthly scales, as shown by Decker et al. (2012) and Wang and Zeng (2012), supporting the finding that temporal resolution can mitigate some spatial resolution discrepancies.

Finally, although the observation data is constrained in coverage, the variables analysed represent the primary hydrometeorological drivers used in land surface modelling, flux estimation, and evapotranspiration calculations. Their evaluation remains directly relevant to hydrological modelling applications, even if spatial representativeness is incomplete. The analysis is centred on these core meteorological variables rather than flux-derived quantities such as latent heat flux or evapotranspiration, which are not evaluated. This focus is deliberate, as gridded evapotranspiration products are themselves modelled outputs that combine meteorological forcing with additional model parameterisations, introducing further layers of uncertainty and would require a separate methodological framework, including validation against flux-derived evapotranspiration based on energy balance closure. By evaluating the fundamental meteorological drivers directly, the study isolates the first-order controls of land-atmosphere exchange and provides a more practical assessment of the input data quality underpinning hydrological and land surface modelling applications. In addition, it paves the way for the development or improvement of other models to calculate evapotranspiration using validated meteorological inputs. Their omission reflects the present study's focus on fundamental drivers of hydrological modelling, while still providing a foundation for examining how errors in meteorological drivers may propagate into derived flux products.

## 6. Implications and Conclusions

This study evaluated five high-resolution meteorological global data products against 11 flux tower observations across Brazil revealing that no single data product consistently performs best across all variables and time scales. However, higher spatial

and temporal resolution products (ERA5-Land and MSWEP) generally outperform the lower resolution counterparts (GLDAS2.0, GLDAS2.1 and BNMD) at the daily scale. As an overall product, the ERA5-Land dataset outperformed the others at both daily and monthly time-steps.

Decomposition of the MSE provided critical insights into the primary sources of error for each variables, underlining correlation error as the most significant contributor for variables with high temporal variability, such as air temperature and precipitation, especially at finer temporal resolutions. This decomposition analysis is instrumental in guiding data product selection for model applications, as it reveals how error sources shift with variable and temporal scale, helping users weigh the importance of bias, variability and correlation error depending on the application goals.

Spanning multiple climatic zones with high-quality observational data across varied time periods, this study offers valuable insights into the robustness and applicability of each data product. The findings support the use of high-resolution reanalysis products, such as ERA5-Land and MSWEP, to enhance model predictive power; however, site-specific validation remains essential for optimal performance before dataset selection. In the absence of observational data or when time constraints limit validation efforts, studies like this that validate gridded datasets across diverse climatic regions become critical.

Moreover, the results emphasise the need for careful consideration of dataset characteristics and application context when selecting a gridded data product. For instance, in applications like evapotranspiration modelling for agriculture, datasets that perform well in the dry season may be preferable (Blankenau et al., 2020). Conversely, for studies assessing long-term ecosystem responses, data products that exhibit stable performance over extended periods may be more suitable (Schymanski et al. 2015). Bias correction methods and data processing steps not covered in this study may further influence dataset

performance, suggesting avenues for future research. This study, alongside others, highlights that cautious, context-specific dataset selection is essential for reliable applications in environmental and climate modelling.

## Data availability

Flux tower observation were provided by the four institutes that took part in the study and data for the early period (2000-2012) was part of the LBA-DIMP (https://www.climatemodeling.org/lba-mip/), while more recent data can be found as supporting material to Melo et al.'s study (2021). Four, gridded products are open access, BNMD Xavier et al. (2016), (https://utexas.app.box.com/v/Xavier-etal-IJOC-DATA), GLDAS2.0 and GLDAS2.1, Rodell et al. (2004), (https://disc.gsfc.nasa.gov/datasets?keywords=GLDAS), ERA5-Land, Muñoz Sabater, J. (2019), (https://cds.climate.copernicus.eu/datasets/reanalysis-era5-land?tab=download). The precipitation dataset MSWEPv2.2 was kindly provided by Hylke Beck over personal contact (Beck et al., 2021).

## Author contribution

JB conceptualised the study, conducted the analysis, and wrote the manuscript. Supervision and guidance were provided by RR and RW. Flux tower data across Brazil were managed and supplied by HR and DR. All authors contributed to the manuscript review and editing.

## Competing Interest

The authors declare that they have no conflict of interest.

## Acknowledgements

The author acknowledges funding to carry out the research from the following funding bodies; The Engineering and Physical Sciences Research Council (EPSRC) Water Informatics: Science and Engineering Centre for Doctoral Training (WISE-CDT; grant no. EP/L016214/1), the Brazilian Experimental datasets for MUlti-Scale interactions in the critical zone under Extreme Drought (BEMUSED; grant no. NE/R004897/1), and partial support from the COSMIC-SWAMP project, jointly funded by NERC under grant NE/W004364/1, and the São Paulo Research Foundation (FAPESP) under grant 2021/03032-7. HR acknowledges financial support, in part of FAPESP 2021/11762-5 and CTG-IAG P&D/Aneel - Sistema inteligente para benefício da qualidade das informações climáticas e ampliação da rede de estações meteorológicas no setor de energia hidro-eólica-solar do Brasil.

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
