# Peer review of "Evaluation of high-resolution meteorological data products using flux tower observations across Brazil"

_EGUsphere, 2025_

## Author Response (AR1)

Author's response

**Reviewer 1**

1.  Selection of Flux Tower Sites

**"*Using only 11 sites seems insufficient for a study area as large and ecologically diverse as Brazil… why limit the analysis to these specific 11 flux tower sites? In my view, excluding conventional meteorological data would make sense only if the primary aim were to assess the ability of gridded products to represent actual ET. However, based on Section 2.1 of the manuscript, "Variables were included so that reference evapotranspiration could be calculated according to the standard FAO methodology." If that is the case, then incorporating data from the Brazilian National Meteorological Institute (INMET) could significantly increase the number of observation points and improve geographic coverage.***

**ic*While it's acknowledged that South America has far fewer flux towers compared to North America or Europe, other flux towers in Brazil—available in the AmeriFlux or FLUXNET networks—are not listed in Tables 1 and 2. I strongly encourage the authors to consider including towers located in underrepresented biomes, such as the Caatinga, Pantanal, and Atlantic Forest. In summary, the authors should elaborate more clearly on why only those 11 flux towers were selected.*"**

We appreciate the importance of this comment. We agree that Brazil's ecological diversity warrants a wide spatial sampling. At the time we began this study, access to high-quality micrometeorological data was limited. The 11 flux tower sites selected represent those for which full meteorological forcing data were curated and available, including seven variables which are the primary hydrometeorological drivers used in land surface models. These variables underpin the estimation of surface energy and water fluxes, allowing for the calculation of evapotranspiration, which in turn provides essential inputs for many hydrological models. We also had access to two extra sites CAX (Caxiuanã, a tropical rainforest riverine in the state of Para) and USE (Usina Santa Eliza, a sugarcane site in the state of Sao Paulo) which were rejected because they had data quality issues or lacked complete or sufficient data coverage to meet the quality control threshold.

Despite omitting some biomes, we feel that the main largest biomes (particularly, the Amazon and the Cerrado) as well as croplands and grasslands/pastures, have been represented. Nevertheless, this omission has been mentioned explicitly in the revised version of the manuscript as a limitation to our study (Section 2.1; line 112-121 and section 5.5; line 500-504), while encouraging other studies to account for these biomes if they wish to do so.

We have also provide a detailed explanation in the revised manuscript on why conventional meteorological stations (INMET) were not included (line 122-126, line 505-509). Importantly, some of the reanalysis and blended products assessed (e.g., BNMD) incorporate INMET station data in their development. Using flux tower data, which are entirely independent of INMET, allows us to evaluate these products more objectively. We highlight the use of FLUXNET towers used as an evaluation metric for MSWEP (Beck et al, 2022, https://doi.org/10.1175/BAMS-D-21-0145.1) where it states *"The FLUXNET data were used for this purpose (evaluate datasets) because they are completely independent; they have not been used in the development of any of the P datasets"* clarifying the importance of independent observations. This has been clarified in the revised version (line 125).

2. Dataset Size and Statistical Robustness

**"Performance metrics derived from larger datasets are generally more reliable, with increased statistical significance and reduced uncertainty. Thus it would be helpful to demonstrate that, despite site differences, the results remain comparable."**

We agree and have carried out a sensitivity analysis using subsets of the longest flux tower time series (e.g., PDG, CRA, K34, K77). This has allowed us to assess how results vary with sample size. We have summarised these results and made them available as supplementary material acknowledging this in the methodology (section 3.8). Performance metrics did not show large any discrepancies.

3. Temporal Averaging and Missing data bias

**"Regarding temporal averaging, it is not clear whether the hourly samples retrieved in each iteration were selected randomly. While the use of two-sample K-S tests is appropriate, its efficiency may vary depending on the time of day during which the data gaps occur. For instance, under a 30-minute resolution, a dataset with evenly distributed missing values (e.g., one every hour) is likely to be much smaller than a sample with missing records only at night, for example.**

**As for the conversion from daily to monthly averages, is using only 50% of the days sufficient? Is there evidence that this threshold does not compromise monthly estimates? For example, if most of the missing days were cloudy, the resulting monthly average could be biased toward sunnier conditions."**

We thank the reviewer for raising this point. To clarify, our analysis was performed at daily and monthly time steps only.

With regards to the infilling, there were obvious failures of instrumentation but in some cases, there was more than one instrument recording similar measurements (e.g.

global radiation in, PAR in and net radiation) these had extremely strong correlations and (in some cases) vastly increased the temporal coverage of a variable.

For conversion to daily means, we used a 50% hourly coverage threshold as a compromise between data availability and temporal representativeness. We tested stricter thresholds (e.g., 100, 90, 80, 70, 60%) and observed a reduction in the number of valid days across sites, with only marginal gains in accuracy (comparing using K-S tests to a reference mean for each variable, whilst also looking at the standard deviation). While we acknowledge the potential for bias (e.g., from systematically missing cloudy days), our sensitivity analysis indicated that the 50% threshold did not significantly affect our daily or monthly estimates. This has now been discussed explicitly in the manuscript (lines 216-225, section 3.2).

4. Spatial Resolution Harmonisation

***"It is unclear why no similar approach was applied to harmonise spatial resolution among the products... I am concerned about the fairness of comparisons involving coarser-resolution datasets."***

We recognise that spatial resolution mismatch can affect the results of point-to-pixel comparisons, particularly for coarse-resolution products. However, we have decided to focus on comparing the temporal performance of different gridded products against independent, high quality point observations, rather than to perform spatial interpolation or downscaling.

Importantly, even sophisticated interpolation efforts do not guarantee spatial consistency in performance. For example, Xavier et al. (2016) found that the accuracy of different interpolation methods varied substantially across Brazil, with no clear spatial or geographic patterns explaining where a given method performed best. Their results highlight the complexity of spatial error structures and suggest that resolution harmonisation may not systematically improve agreement with observations. Therefore, while we acknowledge this is a limitation, spatial harmonisation was not applied, as it could introduce new biases or mask product-specific spatial characteristics.

We have added a discussion of this limitation in the revised manuscript, along with appropriate citations (e.g., Xavier et al., 2016; Hofstra et al., 2008), and clarified that addressing spatial representativeness through interpolation or downscaling is outside the scope of this study (lines 519-527, section 5.5).

We have reviewed the full manuscript and addressed smaller clarity issues noted implicitly in the reviewer's general remarks and specific comments. Furthermore, we have ensured that grammar has improved in clarity.

Reviewer #2

We thank the reviewer for their thoughtful and constructive comments. We are please that the reviewer found the manuscript to be well-written and that the findings align with the stated objectives. We also appreciate the recognition of our use of MSE decomposition and the identification of ERA5-Land as the best performing product overall.

1. Use of INMET data network

*"Since the authors are evaluating gridded meteorological products, a more comprehensive dataset/network, such as the Brazilian National Meteorological Institute (INMET) data, could have been used."*

We appreciate this suggestion and agree that INMET data represent and important observational resource for Brazil. However, we intentionally used flux tower data as an independent evaluation dataset (as explained to similar comments by Reviewer #1) (line 122-126, line 505-509). Some of the reanalysis and blended products evaluated in this study (e.g. BNMD) already assimilate INMET data as part of their development. Using flux tower observations, which are entirely independent from these conventional meteorological networks, provides a more objective evaluations of gridded product performance. This rationale has been clearly articulated in the revised manuscript.

2. Lack of direct comparison between gridded products and flux tower variables

*"Despite using flux tower data, no specific flux tower variables were tested against the gridded products."*

In this study, we focused specifically on core meteorological variables recorded by the flux towers as the basis for comparison with the gridded products. While we did not assess flux-derived variables such as latent heat flux or evapotranspiration, we agree that doing so would offer valuable insights into how differences in meteorological forcing translate to land-atmosphere exchange estimates. We have clarified this scope in the revised manuscript and outlined the evaluation of derived variables such as evapotranspiration as a key direction for future work.

3. Limited geographic coverage of flux tower sites

*"Other flux towers in Brazil could have been used to cover other regions, such as the Northeast, where the climate is predominantly semi-arid."*

We agree that Brazil's ecological diversity warrants a wide spatial sampling (as also responded to similar comments made by Reviewer #1). At the time we began this study, access to high-quality micrometeorological data was limited. The 11 flux tower sites selected represent those for which full meteorological forcing data were curated and available, including seven variables which are the primary hydrometeorological drivers

used in land surface models. These variables underpin the estimation of surface energy and water fluxes, allowing for the calculation of evapotranspiration, which in turn provides essential inputs for many hydrological models (radiation, wind speed, humidity, etc.). We had access to two extra sites CAX (Caxiuanã, a tropical rainforest riverine in the state of Para) and USE (Usina Santa Eliza, a sugarcane site in the state of Sao Paulo) which were rejected because they had data quality issues or lacked complete or sufficient data coverage to meet the quality control threshold.

Despite omitting some biomes, (i.e. Caatinga and Pantanal), we feel that the main largest biomes (particularly, the Amazon and the Cerrado) as well as cropland and grasslands/pastures, have been represented. This omission has been mentioned explicitly in the revised version of the manuscript as a limitation to our study, while encouraging other studies to account for these biomes if they wish to do so.

4. Assessment of evapotranspiration as an integrated variable

   **"*Since evapotranspiration is an important variable from flux towers and models, it would be interesting to test the precision of gridded products against a variable that takes into consideration all the base meteorological data.*"**

   We agree that evapotranspiration is a highly integrative and policy-relevant variable. However, as noted above, the current study focused on the direct evaluation of meteorological forcing variables. A full assessment of ET would require a different methodological framework and validation against flux-derived ET estimates (e.g., via energy balance closure), which we believe to be beyond the scope of our study. We appreciate this suggestion and have explicitly mentioned this point in the discussion as a direction for future work (Section 5.5).

   Specific comments

   **Table 1 - "*Please add the average temperature and precipitation.*"**

   We have revised Table 1 to include the long-term average temperature and precipitation for each site.

   **Lines 165-167 – "*Clarify how the linear gap-filling was applied. Did the authors apply the same methodology for precipitation?*"**

   The gap filling method for precipitation has been explained in a separate paragraph in section 3.2, however, we have elaborated on how the linear regression was performed i.e. what variables were used for the interpolation.

   **Lines 255-256: "*What is the reason why MSWEPv2.2 performed better for daily rainfall BDMB at a monthly timescale?*"**

The way we chose to lay out the manuscript was to first state the results and then discuss them in the following section. An attempt to explain this has been made in line 386 in the manuscript, we have added a caveat "see further explanation in Section 5.2."

---

## Author Response (AR2)

We would like to thank the referee's and the editor for their acceptance subject to minor revisions. We have gone through the manuscript and altered both the minor issues raised by both referee #1 and the editor. Further to this, both JB and RR have gone through and altered some other grammatical errors, improving the overall presentation of the manuscript as per recommendation.